# A Flexible *a*-SiC-Based Neural Interface Utilizing Pyrolyzed-Photoresist Film (C) Active Sites

**DOI:** 10.3390/mi12070821

**Published:** 2021-07-13

**Authors:** Chenyin Feng, Christopher L. Frewin, Md Rubayat-E Tanjil, Richard Everly, Jay Bieber, Ashok Kumar, Michael Cai Wang, Stephen E. Saddow

**Affiliations:** 1Department of Electrical Engineering, University of South Florida, Tampa, FL 33620, USA; chenyinfeng@usf.edu; 2Department of Mechanical Engineering, University of South Florida, Tampa, FL 33620, USA; tanjil1@usf.edu (M.R.-E.T.); kumar@usf.edu (A.K.); mcwang@usf.edu (M.C.W.); 3NeuroNexus LLC, Ann Arbor, MI 48108, USA; cfrewin@neuronexus.com; 4Nanotechnology Research & Education Center, University of South Florida, Tampa, FL 33620, USA; everly@usf.edu (R.E.); bieber@usf.edu (J.B.); 5Department of Medical Engineering, University of South Florida, Tampa, FL 33620, USA

**Keywords:** pyrolyzed-photoresist-film, implantable neural interface, silicon carbide biotechnology, microfabrication, microelectrode array

## Abstract

Carbon containing materials, such as graphene, carbon-nanotubes (CNT), and graphene oxide, have gained prominence as possible electrodes in implantable neural interfaces due to their excellent conductive properties. While carbon is a promising electrochemical interface, many fabrication processes are difficult to perform, leading to issues with large scale device production and overall repeatability. Here we demonstrate that carbon electrodes and traces constructed from pyrolyzed-photoresist-film (PPF) when combined with amorphous silicon carbide (*a-*SiC) insulation could be fabricated with repeatable processes which use tools easily available in most semiconductor facilities. Directly forming PPF on *a*-SiC simplified the fabrication process which eliminates noble metal evaporation/sputtering and lift-off processes on small features. PPF electrodes in oxygenated phosphate buffered solution at pH 7.4 demonstrated excellent electrochemical charge storage capacity (CSC) of 14.16 C/cm^2^, an impedance of 24.8 ± 0.4 kΩ, and phase angle of −35.9 ± 0.6° at 1 kHz with a 1.9 kµm^2^ recording site area.

## 1. Introduction

A comprehensive understanding of electrical activity within the nervous system may significantly help scientists find therapeutic solutions for people possessing limited physical and mental functionality due to the effects of disease or trauma [1]. Since the 1970s, silicon-based materials and noble metals (Pt, Au, etc.) have provided the backbone for the fabrication of microelectrode implantable neural probes (mINP), mainly due to the rapid development of advanced techniques within the semiconductor chip industry [2]. While there are many commercially available mINP products which rely on the incorporation of silicon materials, there are also many mINP incorporating polymeric materials as well [3,4]. However, most mINPs for human use have not been approved to date, mainly due to the observed variability in the long-term reliability of these devices [5]. The issues associated with overall mINP reliability have been attributed to a complex set of interactions involving both biotic and abiotic processes. For instance, the physiological neural environment, which possesses high concentrations of ionic and oxidative species, has been attributed to mINP device failure due to chemical interactions with the insulation material which manifests as physical swelling, cracking, film delamination, or physical corrosion [6,7]. Glial cell encapsulation due to the foreign body response results in isolation of the mINP from the neural environment, and has been attributed as a major aspect for device failure [8]. Developing and investigating new materials and fabrication processes which can overcome these mINP issues remains a focus for our research, especially when considering mINP for long-term use in humans.

We chose amorphous silicon carbide (*a*-SiC) as the conformal insulation for our mINP due to research demonstrating its excellent biocompatibility [9], hemocompatibility, and electrical and physical properties [10,11,12]. While *a*-SiC has demonstrated a wide range of dielectric constants, from 2.7–10, optimization of the stoichiometry and process parameters can be used to steer the final value lower [13,14]. Furthermore, *a*-SiC has demonstrated low leakage current (≤10 pA) after the application of ±5 V potential bias [15]. Unlike many other insulating materials used in mINP, *a*-SiC possesses nearly complete chemical inertness, protecting it from oxidative species and ionic diffusion [15,16,17,18].

The electrode material must be suitable for both positive biological interactions as well as long-term electrical performance reliability. As one of the most versatile elements in the periodic table, carbon has been incorporated into many device applications [19]. Since 2000, carbon-based materials, such as carbon nanotubes (CNT) [20], graphene [21], glassy carbon (GC) [22] and pyrolyzed photoresist film (PPF) [23], have been prominently featured in biomedical device applications and show great potential for mINP applications [24,25]. There are debates on the neurotoxicity of CNT and graphene nanomaterials [26,27]. In addition, the fabrication processes for graphene and CNT-based mINPs have been demonstrated as overly complex, potentially expensive, and hard to scale reliably. Furthermore, comparing with GC, PPF has a smoother surface and better electrochemical performance [22]. Thus, in this report, the performance of PPF as a conductive trace, fully encapsulated in *a*-SiC, was evaluated for mINP electrode applications. PPF is synthesized from standard photoresist, a patternable polymeric material containing a photoactive resin dissolved in a solvent. Photoresist thickness and uniformity can be precisely controlled using semiconductor fabrication technology which facilitates highly reproducible processes. PPF has demonstrated excellent biocompatibility [24,28,29], electrical conductivity, stability, and chemical inertness [24,30]. Thus, the development of a carbon-based mINP was undertaken in this work using PPF as the carbon conductive layer with conformal *a*-SiC as the insulation material. Compared to conventional noble metal-based mINPs, PPF can be directly patterned and pyrolyzed from photoresist, which eliminates the need for noble metal evaporation/sputtering and lift-off of small features thus simplifying the fabrication process [31]. In addition, compared with other mINPs based on pyrolyzed carbon materials on polymer substrates [32], in this work PPF is directly formed on a substrate without the need of a transfer step which not only avoids potential damage to the electrodes but also simplifies the manufacturing process [33].

In this paper, we present PPF-based mINP devices with both 16 and 32 channels, as well as planar single-ended PPF electrodes for electrochemical testing. The mINP is formed by sandwiching PPF traces between two layers of *a*-SiC insulation, with windows of 25 µm in diameter for the mINP, and 50 µm to 800 µm in diameter for single ended electrodes to expose the PPF electrodes to the electrochemical environment. A 50 µm thick Si layer serves as a physical support for the thin insulation layers. Each PPF trace is 10 µm wide, ~2 mm long, and 387 nm to 599 nm thick due to variations in the pyrolyzation annealing temperature (i.e., the higher the annealing temperature the thinner the PPF layer). We present the results of physical and electrochemical characterization to motivate further development of PPF-based carbon mINPs or long-term human use.

## 2. Materials and Methods

### 2.1. Device Fabrication Process

Using variations of standard semiconductor micromachining processes, *a*-SiC supported PPF mINP devices were fabricated. The process started with the deposition of a 250 nm thin film of *a*-SiC on a Si wafer by Plasma Enhanced Chemical Vapor Deposition (PECVD, PlasmaTherm PT-700). The deposition process parameters were as follows: substrate temperature 250 °C and chamber pressure 1100 mTorr with 200 W of RF power. The precursors were 360 sccm of CH_4_ and 12 sccm SiH_4_ diluted in 700 sccm of Ar. Figure 1 shows the PPF neural probe fabrication process sequence. The wafer was then diced using a diamond wafering saw (Dicing saw M1006, Micro Automation Inc., Billerica, MA, USA) and cleaned using solvent, followed by an RCA clean. AZ12xt positive photoresist was used to fabricate the device’s conductive traces (i.e., PPF electrodes), since AZ12xt can provide PPF traces in the desired thickness range of 6 µm to 12 µm. AZ12xt was spin-coated on the *a*-SiC-coated wafer at 500 rpm for 20 s, 3000 rpm for 50 s, and 11,000 rpm for 2 s. The first spin stage was to distribute the photoresist on the wafer uniformly, the second stage set the thickness of the photoresist, in this case ~7µm as measured by a stylus profilometer, and the last stage was used to eliminate the edge bead. The coated substrate was then soft-baked at 120 °C for 3 min, followed by UV exposure at an intensity of ~120 mJ/cm^2^. Next, a one-minute post-exposure bake at 95 °C was performed. Pyrolyzation was performed by annealing the photoresist in a tube furnace at atmospheric pressure and at different temperatures from 500 °C to 900 °C under an Ar and H_2_ environment of 100 sccm and 5 sccm, respectively. The thermal ramp rate was 10 °C/min, followed by a thermal soak at the desired temperature for one hour. After annealing, the sample was cooled down to room temperature under the same Ar and H_2_ flow.

The top conformal *a*-SiC insulator layer was deposited on the PPF layers using identical PECVD conditions to the base layer. Contact and electrode windows were opened in the top insulator through reactive ion etching (RIE) employing the following parameters: 37 sccm CF_4_ and 13 sccm O_2_, RF power 200 W, and a base pressure of 50 mTorr for 5 min time. The etch rate was 0.9 nm/s; thus, a timed etch was used and complete removal of the *a*-SiC layer confirmed after this step was completed using a stylus profilometer. In future work an etch stop will be employed to ensure window opening is consistent from batch to batch.

Bond pads were formed on the contact mesas via E-beam evaporation starting with 50 nm of Ti as an adhesion layer followed by 300 nm of Au to facilitate device packaging. The die was then bonded to a Si handle wafer device-side down using crystal bond (Ted Pella, Inc., Redding, CA, USA) and the silicon substrate was thinned using deep reactive ion etching (DRIE) to achieve a 50 µm Si supporting layer. The final process step was to harvest the probes from the Si handle wafer via dissolution of the crystal bond adhesion layer in acetone. The influence of the electrode geometric area on PPF performance was investigated through single-ended electrodes with variable recording areas. In this case we did not thin the Si substrate since these devices were not to be implanted. Electrode areas investigated are 502, 125, 7.8, 1.9 and 0.49 kµm^2^, respectively. In order to directly compare the performance of the PPF devices with mINP based on noble metals, a control group of Pt single-ended electrodes were fabricated using the same procedure, with Pt replacing the PPF layer.

### 2.2. Surface Characterization of a-SiC Insulated PPF Probe

Surface characterization of both the *a*-SiC base layer and PPF conductive traces were conducted by scanning electron microscopy (SEM), Raman spectroscopy, and X-ray Photoelectron Spectroscopy (XPS, Physical Electronics Industries 5400 LS). To verify that PPF traces and electrodes were formed during annealing, Raman Spectroscopy (NRS-4500, JASCO, Inc., Easton, MD, USA) was used with a 20 mW green laser (532 nm). Each data point was collected during a 5 sec exposure time and averaged over 15 exposures. The *a*-SiC stoichiometry and oxygen concentration were evaluated as a function of depth via XPS. XPS depth profiling was performed on *a*-SiC with a raster size of 5 × 5 mm with a beam current of ~1 µA. The morphology of both the PPF probe and single-ended electrodes were examined by SEM.

### 2.3. Electrical and Electrochemical Characterization of a-SiC Insulated PPF mINP

Double-ended resistors were used to characterize the electrical properties of the PPF material. The current-voltage (I/V) curves were measured from these devices, and the PPF resistivity calculated from the measured resistance and electrode cross-sectional area and length. Characterization of *a*-SiC insulation was performed using interdigitated electrodes (IDE) tested by current-voltage (IV) measurement between the range of −50 V to +50 V under both dry and wet (PBS) environments. The IDE device consists of finger electrode traces with two parallel traces with a width of 50 µm, a length of 1 mm, separated by 100 µm, and containing 22 digits. A Keithley 2400 was used to apply the voltage on the long trace and the leakage current was evaluated by recording the current on the finger electrode traces. Electrochemical characterization was performed via cyclic voltammetry (CV) and electrochemical impedance spectroscopy (EIS) using a potentiostat (Reference 600, Gamry, Inc., Philadelphia, PA, USA) in the three-electrode measurement configuration as follows: working electrode (PPF), counter electrode (Pt), and reference electrode (Ag/AgCl). The charge storage capacity (CSC) was calculated from the CV data. The measurements were performed in N_2_ aerated pH balanced (7.4) phosphate buffered saline (PBS) at room temperature. The frequency range of the EIS measurements was from 10 Hz to 100 kHz, and 10 data points were collected per decade. CV measurements were performed within a safe potential window (determined experimentally) between −0.7 to 1.5 V at a scan rate of 50 mV/s with three complete repetitions.

## 3. Results

### 3.1. Surface Characterization

The *a*-SiC supported PPF neural probe was fabricated by the method described above. In order to determine the optimum annealing temperature, PPF traces were annealed from 500 °C to 900 °C in 100 °C increments and characterized. In order to accurately measure the thickness of each layer of the device, cross-section SEM was used as shown in Figure 2a (PPF trace annealed at 900 °C). In this figure, we can observe that the photoresist-film thickness decreased from 7 µm to 0.387 µm after pyrolyzation. The total vertical thickness shrinkage was ~93–95% with negligible lateral shrinkage. Combined with the *a*-SiC base layer and thinned Si substrate and capping layer, the total thickness of our functional stack was ~51 µm in total. Figure 2b shows an optical micrograph of two PPF/*a*-SiC probes, one single-shank (16 channel) and one double-shank (32 channel), after probe release. As described earlier there is a 50 µm Si layer on the backside which serves as a support layer. The devices are 5 mm in length, 2 mm in width, and ~51 µm in thickness. Figure 2c shows a planar SEM micrograph of the 16 channel PPF neural probe shank from Figure 2b. The recording area of each tip was 314 µm^2^ (inset), and the width of each trace was 10 µm with a total shank width of 300 µm. XPS depth analysis on as-deposited *a-*SiC, high-resolution C 1s and Si 2p, are shown in Figure 3a–c. The depth profile was obtained via Ar^+^ Milling with a beam energy of 3.5 kV, 5 × 5 mm^2^ raster size and an X-Ray power of 350 W. Both the Ar^+^ milling and XPS analysis were conducted at a base pressure of 3 × 10^−10^ Torr. The depth profile illustrates that the oxygen content was mainly in the near-surface region of the *a-*SiC film, and it decreases significantly within ~15 nm. Furthermore, Figure 3b,c and Table 1 indicate that *a*-SiC was deposited, which has a low oxygen content of 8.53% and C:Si ratio of 0.9:1.

### 3.2. Electrochemical and Electrical Characterization

Two double-ended resistors were used to characterize the electrical properties of the PPF electrodes. Each PPF resistor had a length of 3 mm, a width of 50 µm, and the thickness varied depending on the annealing temperature. Table 2 shows a comparison of PPF resistivity and thickness vs. annealing temperature. The results show that the resistivity decreases with increasing annealing temperature. Figure 4 is the IV curve of the IDE device under both dry and wet (PBS) conditions. The *a*-SiC insulation leakage currents at ±50 V were less than 7 nA and 19 nA dry and wet (PBS), respectfully. The electrochemical properties of both PPF and control Pt electrodes were examined by EIS and CV. PPF electrodes annealed at 900 °C showed the lowest impedance at 1 kHz comparing with PPF annealed at lower temperature, therefore 900 °C was chosen for subsequent electrochemical characterization. The measurements were performed in nitrogen-aerated 7.4 pH PBS and the results are shown in Figure 5. CV curves for the PPF and Pt electrode with different recording site areas are shown in Figure 5b,c. Figure 5d shows CV curves from the 900 °C PPF vs. Pt control electrode with the same recording area of 502 kµm^2^, using a Pt electrochemical window (−0.6 V to +0.8 V). The anodic charge storage capacity (CSC_a_) and cathodic charge storage capacity (CSC_c_) were calculated by taking the time integral of the anodic current and cathodic current, respectively. The CSC_a_ and CSC_c_ for the 900 °C PPF electrode with 502 kµm^2^ area were 15.06 mC/cm^2^ and 18.07 mC/cm^2^ which is significantly higher than 3.9 mC/cm^2^ and 5.2 mC/cm^2^ of the reference Pt electrode with same recording area. The EIS results for the same PPF electrodes are shown in Figure 5a, while the solid line represents PPF results and the dashed line represents Pt results. The phase diagram and CV curves show that the PPF electrodes displayed a shift to a more resistive, faradaic current centered at 1 kHz. The impedance value at 1 kHz is normally associated with the frequency of the action potentials of active neurons. The impedance amplitude for our PPF electrode annealed at 900 °C with 490 µm^2^ area was 129.6 ± 25.7 kΩ at 1 kHz, which is 71% lower than that of the Pt reference electrode (437.8 ± 35.9 kΩ) with the same recording area. This result indicates that the PPF on *a*-SiC electrode appears to have an excellent impedance range for recording neural action potentials.

## 4. Discussion

A novel mINP constructed via *a*-SiC insulated PPF traces was fabricated mimicking silicon electrode planar architecture along with a Pt control group for direct comparison. The PPF-mINP was characterized using EIS and CV electrochemical measurement and demonstrated promising performance with an empirically determined potential window of −0.7 V to 1.5 V, a low impedance (24.8 ± 0.4 kΩ) at 1 kHz, a phase angle of −35.9 ± 0.6° with a CSC_a_ of 8.9 C/cm^2^ and a CSC_c_ of 5.26 C/cm^2^ with 1.9 kµm^2^ recording site area. For comparison, the Pt control group had an impedance of 103.8 ± 2.2 kΩ at 1 kHz and a phase angle of −94.5 ± 2.8° with CSC_a_ of 4.7 mC/cm^2^ and a CSC_c_ of 7.4 mC/cm^2^.

A combination of methane (CH_4_) and silane (SiH_4_) were used as precursors to PECVD deposit *a*-SiC. The mean roughness of as-deposited *a*-SiC film was approximately 1 nm on both the center and edge parts of the wafer which indicates a smooth *a*-SiC surface which is important for a precise photolithography process. Figure 2a,b show SEM images of the probe’s cross-section view and top view. In Figure 2a, the cross-section view of PPF on top of *a*-SiC was shown with a PPF thickness of 387 nm. Prior to annealing, the AZ12xt photoresist had a thickness of 7 µm, indicating a thickness reduction during annealing of ~93–95%. However, there was virtually no change in the lateral dimension during the annealing. While the ambient temperature was ramping up during annealing, the solvent phenols in the polymer evaporated before the sample temperature reached 150 °C. After 300 °C, C=O bonds will be broken leading to C-C bonds and, ultimately, C=C bonds are formed, with the very bottom C layer bonding with the *a*-SiC surface [24]. However, during annealing the bottom atoms were bonded with the dangling bonds at *a*-SiC surface which prevented it shrinking in lateral size. Thus, the vertical dimension is expected to shrink while the lateral dimension would not experience appreciable change. Raman spectra have confirmed the sp^2^ hybridized carbon formed by high-temperature annealing. Thanks to the significant thickness reduction, we were able to fabricate ultrathin PPF mINPs, which is important to maintain surface planarity for subsequent device fabrication processes. A top view SEM image on our PPF mINP was shown in Figure 2c. A 16 channel mINP was demonstrated with a 314 µm^2^ recording site area. In Figure 2b, an overview of the *a*-SiC/PPF/*a*-SiC neural probes was shown.

XPS depth analysis and high-resolution spectra of the base *a*-SiC film were shown in Figure 3. The oxygen content inside the *a*-SiC film was determined by XPS depth profiling as shown in Figure 3a. Argon sputtering was using to raster mill the *a*-SiC surface within a 5 mm^2^ area for 24 cycles, with a sputtering rate of approximately 10 nm per cycle. XPS data was collected after each cycle so that concentration vs. depth information could be obtained. A very low oxygen concentration inside the *a*-SiC film is crucial to ensure that the film is indeed *a*-SiC and not a carbon-doped oxide layer, which would be water soluble Si-O-C instead of robust *a*-SiC. Before the first cycle, the oxygen content was much higher than the silicon and carbon content. Within three sputter cycles, the oxygen content dramatically dropped to almost zero, thus illustrating that the observed oxygen atoms were only on the surface, likely due to contamination from the ambient air or a minute amount of surface oxide. Figure 3b indicates that Si-C bonds were formed, demonstrating that *a*-SiC was indeed deposited. Table 1 showed C:Si ratios was 0.9.

It is known that *a*-SiC film deposited by PECVD will result in a high film stress [34], and small film stress is desired for neural probes application to obtain a straight probe’s shank after been released from the Si substrate. Thus, optimization of film stress is necessary. In this work, the curvatures of *a*-SiC on Si before and after thermal annealing were measured by Dektak D150, then film stresses were calculated by modified Stoney’s equation. The as deposited 250 nm *a*-SiC had a compressive film stress of 250 MPa, and after annealing at 900 °C for 1 hr, the film stress was determined to be 200 MPa tensile stress. This result indicated that thermal annealing can release the film stress.

Since the PPF annealed at 900 °C showed the lowest impedance results compared with lower temperature annealing, we will focus our electrochemical characterization discussion on the 900 °C PPF electrodes. The electrochemical properties of both PPF and Pt reference single-ended electrodes vs. recording areas was measured using a Gamry Reference 600 potentiostat/galvanostat (Gamry Instruments, Warminster, PA, USA). In this research, results from single-ended electrodes with five different recording areas (0.49, 1.9, 7.8, 125, and 502 kµm^2^) were shown in Figure 5. Figure 5a showed that, at 1 kHz, the impedance was 125.6 ± 25.7 kΩ for the PPF electrode with a recording area of 0.49 kµm^2^ which is 71% lower than that for the Pt reference electrode (437.8 ± 35.9 kΩ) with the same recording area. In addition, with increasing recording area, the impedance of the PPF electrode decreases, as expected. The PPF electrodes with 1.9 kµm^2^ area had an impedance of 24.8 ± 0.4 kΩ compared with 103.8 ± 2.2 kΩ for the reference Pt electrode. However, the phase angles (−35.9 ± 0.6°) from EIS measurements at 1 kHz and indicate both faradaic and capacitive charge transfer mechanisms at the electrode-electrolyte interface which was consistent with carbon-based materials from other works [30,35,36], this indicate that carbon was oxidized and reduced at the surface of the electrode to transfer electron flow from the electrode to ions flow in the electrolyte [37]. This differs from the phase angle results of the Pt control group (−94.5 ± 2.8°), which present a dominant capacitive charge transfer mechanism. The CV measurements on the smallest electrodes (0.49 kµm^2^) showed a significant noise signature and, as a result, are not reliable, thus we will focus on the larger area electrodes hereafter. One important characteristic was calculated from CV curves which is charge storage capacity, it illustrates the total potential charge available for inject of the electrode during stimulation [38]. Table 3 has summarized the PPF and Pt electrodes EIS data at 1 kHz as well as the anodic charge storage capacity (CSC_a_) and cathodic charge storage capacity (CSC_c_) from the CV data. The 1.9 kµm^2^ PPF electrodes showed excellent CSC_a_ and CSC_C_ of 8.9 C/cm^2^ and 5.3 C/cm^2^, respectfully, with a scanning range of −0.7 V to +1.5 V. The larger CSC_a_ and smaller CSC_c_ value was caused by the anodic oxidation of carbon on the PPF electrodes. In Figure 5d and Table 3, compared with Pt within the range of −0.6 V to +0.8 V, the 502 kµm^2^ PPF showed CSC_a_ and CSC_c_ of 11.88 mC/cm^2^ and 16.08 mC/cm^2^, respectfully, while Pt showed CSC_a_ and CSC_c_ of 3.9 mC/cm^2^ and 5.2 mC/cm^2^, respectively. This result indicates our PPF electrodes have ~4 times higher CSC than Pt electrodes of same recording area, a very significant result. The large CSC of PPF electrodes were attributed to the electron transfer through Faradaic chemical reaction mechanisms. The excellent electrochemical properties of the PPF electrodes indicates that it is a promising candidate for mINP applications. In Table 4, the electrochemical impedance and charge storage capacity of various commonly used mINP materials are shown. In this framework, the PPF electrodes showed good impedance at 1 kHz and the largest CSC among the other materials, which indicates that PPF is worth further development for next-generation mINPs. In Figure 6, a final device packaged with commercial 16-pin style package (NeuroNexus) is shown. This device is ready for in vitro and in vivo testing and will be reported in a subsequent report.

## 5. Conclusions

Silicon-based neural interfaces are commonly used in present-day neural interface technology. However, silicon and noble metals continue to suffer from long-term reliability issues due to their poor biocompatibility and material degradation, resulting in scar formation and a loss of functionality during recording and stimulation when chronically implanted [43]. These drawbacks have slowed their adoption in medical applications for humans. A novel *a*-SiC/PPF-based neural probe is being developed to solve these challenges and preliminary results have been presented. The PPF probe was fabricated by pyrolyzing photoresist on an *a*-SiC layer deposited on a Si wafer and then conformally insulated using an *a*-SiC capping layer, directly patterning and forming the conducting traces and electrodes, eliminating the noble metal evaporation/sputter and lift-off on small features, and simplifying the fabrication process. The *a*-SiC film showed very low oxygen incorporation via XPS and the films withstood thermal annealing up to 900 °C. The PPF neural probes have 16/32 conducting traces and recording sites that were patterned on the *a*-SiC layer via standard microfabrication processes and are awaiting packaging for continued testing and implantation into an animal model.

Planar single-ended electrodes with different recording areas were fabricated using an identical process to the mINP devices and their electrochemical properties fully characterized and compared to Pt reference electrodes of identical dimension. The surface morphology of the PPF traces was smooth and pin-hole free, while Raman spectroscopy indicated that sp^2^ hybridized carbon film was formed by high-temperature annealing. The electrochemical impedance at 1 kHz of our 1.9 kµm^2^ PPF electrodes was 24.8 ± 0.4 kΩ while that of a Pt control electrode of same recording area was 103.8 ± 2.2 kΩ, indicating the PPF has a 71% lower impedance than Pt on an *a*-SiC support. Moreover, the PPF electrode showed a wider electrochemical window (−0.7 V to 1.5 V) than Pt (−0.6 V to 0.8 V) and 286% larger anodic and cathodic charge storage capacity (CSC), again for the same recording area. These results illustrate the feasibility of neural recording and stimulation with the fabricated *a-*SiC/PPF neural probes. Moreover, the *a*-SiC/PPF mINP is a promising candidate to replace conventional noble metal-based mINPs. While this report focuses on the fabrication and characterization of the *a*-SiC/PPF mINP, we have used these methods to construct a complete implantable device that can be evaluated for long-term stability in vivo, and will be reported at a later date.

## 6. Patents

USPO Provisional Patent serial number 63/199,874 “Pyrolyzed-Photoresist-Film (PPF) on Amorphous Silicon Carbide as a Flexible Chronic Implantable Neural Interface” filed 28 January 2021.

## Figures and Tables

**Figure 1 micromachines-12-00821-f001:**
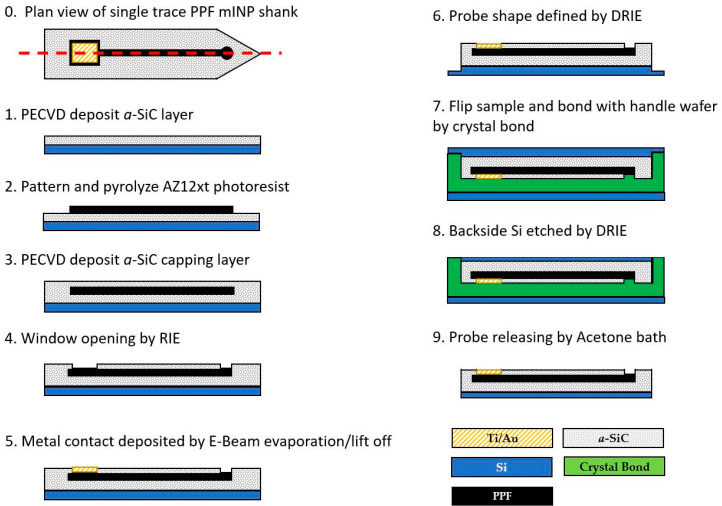
Sketch of a single trace PPF mINP shank for process illustration purposes. (**0**) Top view of completed device; dashed line shows cross-section cut for (**1**–**9**). (**1**) Starting with an oxide-coated Si wafer, an *a*-SiC film is deposited via PECVD. (**2**) Positive photoresist is spun, patterned and thermally annealed to pyrolyze the photoresist into PPF carbon traces. (**3**) Wafer is coated with another layer of *a*-SiC. (**4**) RIE is used to open windows exposing the recording tip and bonding pad. (**5**) Metal is deposited on the bond-pads using a lift-off process and (**6**) probe is defined by DRIE (mINP only, singled ended electrode process completed at step (**5**)). (**7**) Die is flipped and bonded to Si handle wafer using crystal bond. (**8**) Backside Si is thinned to ~50 µm by DRIE. (**9**) Probes are released from Si handle wafer via crystal bond dissolution in Acetone.

**Figure 2 micromachines-12-00821-f002:**
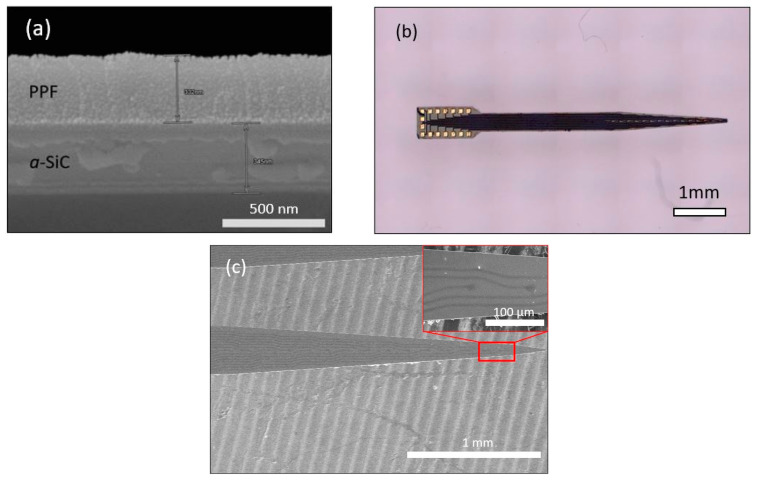
Fabricated PPF on *a*-SiC probe SEM and optical micrographs: (**a**) SEM cross section view after PPF annealing (900 °C). (**b**) Optical image of single shank (top) *a-*SiC/PPF neural probe. (**c**) SEM plan view (inset image showing detail of recording tip).

**Figure 3 micromachines-12-00821-f003:**
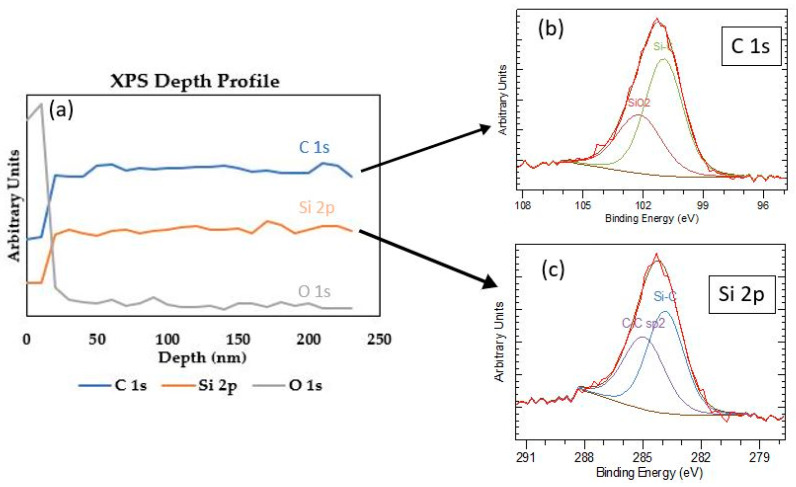
XPS analysis on 300 nm thick *a*-SiC film deposited on a Si wafer. (**a**) XPS depth profile via Ar^+^ milling. The concentration of oxygen is below the detection limit at ~15 nm depth indicating no oxygen in the *a*-SiC bulk. High-resolution (**b**) C 1s and (**c**) Si 2p peaks at binding energies indicative of Si-C bond formation pre-anneal. XPS survey on *a*-SiC.

**Figure 4 micromachines-12-00821-f004:**
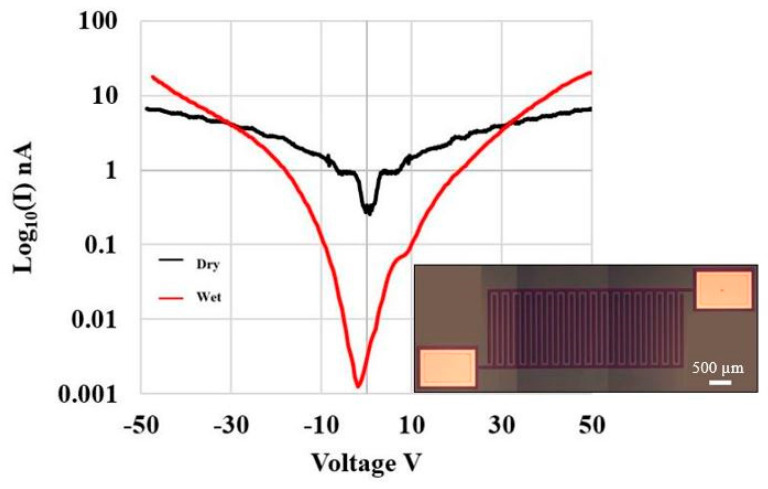
Electric field breakdown data used to evaluate *a*-SiC insulation properties via IV measurement of leakage current from a PPF/*a*-SiC IDE device. Two-die were tested under dry/wet (PBS) conditions and the average leakage current at ±50 V was 7 nA (dry) and 19 nA (wet). Photograph of IDE device (inset) with 100 µm spacing between PPF mesas (scale bar 500 µm).

**Figure 5 micromachines-12-00821-f005:**
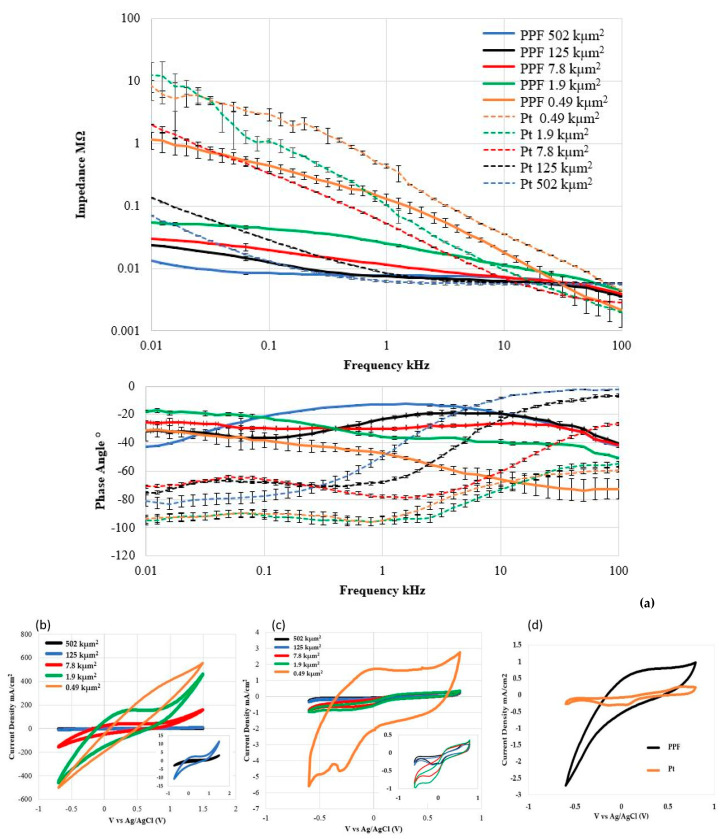
PPF and Pt reference electrode electrochemical characterization data. (**a**) PPF and. Pt EIS impedance vs. frequency, and CV curves vs. recording area for (**b**) PPF and (**c**) Pt traces. (**d**) CV data comparison of PPF (900 °C) vs. Pt with same recording area (502 kµm^2^). Note significantly larger curve area for PPF vs. Pt indicating superior charge storage capacity.

**Figure 6 micromachines-12-00821-f006:**
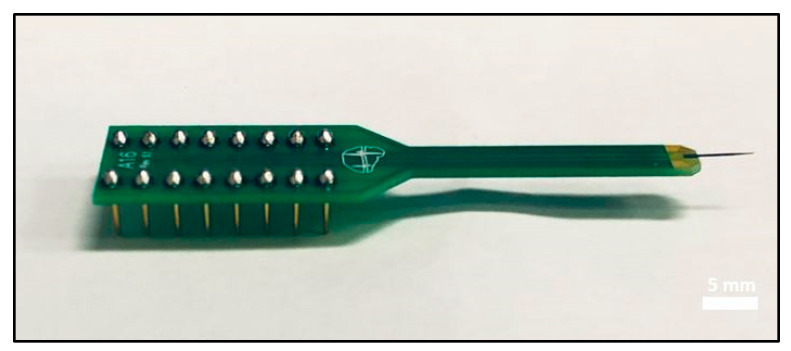
Final packaged 16-channel PPF device on a commercial probe header (NeuroNexus).

**Table 1 micromachines-12-00821-t001:** XPS *a*-SiC film atomic concentration pre- and post- anneal.

Species	Atomic Concentration
Carbon	43.10
Silicon	48.37
Oxygen	8.53
C:Si ratio	0.9

**Table 2 micromachines-12-00821-t002:** Measured PPF resistivity vs. annealing temperature using double-ended resistor mesas *.

Annealing Temperature (°C)	PPF Thickness (nm)	Resistivity mΩ∙cm)
500	~599	24
600	~498	11
700	~459	8
800	~424	4
900	~387	3

* mesa width and length, 50 µm and 3 mm, respectively.

**Table 3 micromachines-12-00821-t003:** The impedance at 1 kHz of 900 °C PPF vs. Pt.

Electrode *	Area (kµm^2^)	Impedance 1 kHz (kΩ)	CSC_a_ (mC/cm^2^)	CSC_c_ (mC/cm^2^)
PPF	0.49	125.6 ± 25.7	5600	3000
Pt	0.49	437.8 ± 35.9	53.2	33.2
PPF	1.9	24.8 ± 0.4	8900	5260
Pt	1.9	103.8 ± 2.2	4.7	7.4
PPF	7.8	11.4 ± 0.04	2870	1600
Pt	7.8	51.8 ± 1.1	4.9	6.4
PPF	125	7.4 ± 0.04	193	107
Pt	125	8.4 ± 0.2	4.1	5.6
PPF	502	7.7 ± 0.01	50	27
Pt	502	6.3 ± 0.1	3.9	5.2

* Electrochemical window for PPF and Pt −0.7 V to1.5 V and −0.6 V to 0.8 V, respectfully.

**Table 4 micromachines-12-00821-t004:** Comparison of electrochemical properties of PPF and common mINP electrode materials.

Material	Recording Area (kµm^2^)	Impedance @1kHz (kΩ)	CSC * (mC/cm^2^)
PPF/*a*-SiC	1.9	24.8	14,160
Pt	1.9	103.8	12
PEDOT/CNT [39]	2.83	15.0	6
Carbon-nanotube fiber [40]	1.450	14.1	372
Graphene Fiber [41]	0.749	37.9	798
IrO_x_ [42]	0.177	132.9	29
TiN [39]	2.83	54.8	5

* Charge Storage Capacity (CSC) value sum of anodic and cathodic phases.

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
