# Peer review of "A Flexible a-SiC-Based Neural Interface Utilizing Pyrolyzed-Photoresist Film (C) Active Sites"

_micromachines, 2021, doi:10.3390/mi12070821_

Round 1

Reviewer 1 Report

This paper presents the fabrication, physical and electrochemical characterization of a-SiC/PPF mINP devices. Device fabrication process, surface characterization, electrical and electrochemical characterization were discussed in details. The results of physical and electrochemical characterization this paper presents could establish a foundation for further research/development of PPF-based mINP devices, so this paper should be of interest to the readers in the relevant field. There are some minor suggestions to improve the paper.

1. In the Introduction section, the authors could briefly address the potential performance of the fabricated devices in comparison with the conventional ones.

2. The Introduction section will be also strengthened if the authors could review other research groups' PPF-based mINP studies. Why are their fabricated devices better or comparable to others? 

3. More details on the analysis of Figure 3 and Table 1 are needed in Section 3.1. 

4. The Conclusions section will be strengthened if the authors could address the impacts of their work on the field and the biotechnology society. 

Author Response

Dear reviewer

Thank you for your comments! 

For your comments 1&2, we have added some details in the Introduction “Compared to conventional noble metal based mINPs, PPF can be directly patterned and pyrolyzed from photoresist, which eliminates the need for noble metal evaporation/sputtering and lift-off of small features thus simplifying the fabrication process. In addition, compared with other mINPs based on pyrolyzed carbon materials on polymer substrates, in this work PPF is directly formed on a substrate without the need of a transfer step which not only avoids potential damage to the electrodes but also simplifies the manufacturing process”. 

For your comment 3, we have added details in section 3.1. “The depth profile illustrates that the oxygen content was mainly in the near-surface region of the a-SiC film, and it decreases significantly within ~15 nm. Furthermore, Figure 3(b)(c) and Table 1 indicate that a-SiC was deposited, which has a low oxygen content of 8.53% and C:Si ratio of 0.9:1.”

For your comment 4, we have added a sentence in the conclusion. “These results illustrate the feasibility of neural recording and stimulation with the fabricated a-SiC/PPF neural probes. Moreover, the a-SiC/PPF mINP is a promising candidate to replace conventional noble metal based mINPs.”

Thank you very much!

Best wishes,

Chenyin

Reviewer 2 Report

The paper "A Flexible a-SiC-based Neural Interface Utilizing Pyrolyzed-Photoresist Film (C) Active Sites Chenyin" present the demonstration 
that carbon electrodes and traces constructed from pyrolyzed-photoresist-film (PPF) when combined with  amorphous silicon carbide (a-SiC) insulation could be fabricated with repeatable processes which use tools easily available in most semiconductor facilities.

Some sugestion:

1. The abstract must be improved and presented the experiment more detailed.

2.The keywords can be improved

3.The refernces can be updated with the paper research in the last 5 years.

4.The references cited on the text must chenged ([15] [16][17][18]) like [15-18].

5.The figures are not cleared must be redesigned.

Author Response

Dear reviewer

Thank you for your comments!

For your comment 1: we have added some experimental details in the abstract. “Here we demonstrate that carbon electrodes and traces constructed from pyrolyzed-photoresist-film (PPF) when combined with amorphous silicon carbide (a-SiC) insulation could be fabricated with repeatable processes which use tools easily available in most semiconductor facilities. Directly forming PPF on a-SiC simplified the fabrication process which eliminates noble metal evaporation/sputtering and lift-off processes on small features.”

For your comment 2: we have added more keywords including “microfabrication; microelectrode array”

For your comment 3: we have added several references which conducted the last 5 years

For your comment 4: The in-text citations have been updated to [10-12] and [15-18]. 

For your comment 5: we have updated figures with higher quality. 

Again, thank you very much for your time!

Best wishes,

Chenyin 

Round 2

Reviewer 2 Report

This paper can be published in the present form